# Effect of Dietary Linoleic Acid (18:2n-6) Supplementation on the Growth Performance, Fatty Acid Profile, and Lipid Metabolism Enzyme Activities of Coho Salmon (*Oncorhynchus kisutch*) Alevins

**DOI:** 10.3390/ani12192631

**Published:** 2022-09-30

**Authors:** Hairui Yu, Lingyao Li, Leyong Yu, Congmei Xu, Jiayi Zhang, Xiangyi Qiu, Yijing Zhang, Lingling Shan

**Affiliations:** 1Key Laboratory of Biochemistry and Molecular Biology in Universities of Shandong (Weifang University), Weifang Key Laboratory of Coho Salmon Culturing Facility Engineering, Institute of Modern Facility Fisheries, College of Biology and Oceanography, Weifang University, Weifang 261061, China; 2Shandong Collaborative Innovation Center of Coho Salmon Health Culture Engineering Technology, Shandong Conqueren Marine Technology Co., Ltd., Weifang 261108, China

**Keywords:** fatty acid profile, hepatic enzymes, linoleic acid requirement, *Oncorhynchus kisutch* alevin, proximate composition

## Abstract

**Simple Summary:**

In China, coho salmon is becoming an increasing farming species in recent years. Thus, full understanding of the fatty acid requirements of coho salmon is particularly relevant, especially during the larval stage, since they need high-energy demands and rapid lipid metabolism. As a vital essential fatty acid, linoleic acid (18:2n-6) also plays an important role in the larval stage due to the production of longer-chained and more essential highly unsaturated fatty acids (HUFA). However, there is a lack of knowledge about the dietary linoleic acid (18:2n-6) requirement for this fish species. In this study, six isonitrogenous and isolipidic diets were formulated to contain different dietary linoleic acids (18:2n-6) by increasing corn oil at the expense of coconut oil to study the growth performance, fatty acid profile, and hepatic lipid metabolism enzymes of coho salmon alevins. The results showed that the dietary linoleic acid (18:2n-6) affected the growth performance, muscle fatty acid profile, and hepatic lipid metabolism enzyme activities of coho salmon, and determined the optimal demand of linoleic acid based on the specific growth rate (SGR) and feed conversion ratio (FCR).

**Abstract:**

A 12-week feeding trial aimed to evaluate the effects of dietary linoleic acid (LA, 18:2n-6) on the growth performance, fatty acid profile, and lipid metabolism enzyme activities of coho salmon (*Oncorhynchus kisutch*) alevins. Six experimental diets (47% crude protein and 15% crude lipid) were formulated to contain graded LA levels of 0.11%, 0.74%, 1.37%, 2.00%, 2.63%, and 3.26%. Each diet was fed to triplicate groups of 50 alevins with an initial body weight of 0.364 ± 0.002 g, which were randomly assigned to 18 white plastic tanks (0.8 × 0.6 × 0.6 m, 240 L/tank). Fish were reared in a freshwater flow-through rearing system and fed to apparent satiation four times daily. The survival rate was not significantly different among the treatments (*p* > 0.05). However, the 1.37% LA group significantly improved the final body weight and specific growth rate (SGR) (*p* < 0.05) of alevins. The feed conversion ratio (FCR) in the 1.37% LA group was significantly lower than those in other groups (*p* < 0.05). The whole-body lipid content significantly decreased (*p* < 0.05) with dietary LA levels increasing from 0.74% to 2.00%. The fatty acid composition of the total lipid in muscle was closely correlated with those in the diets. The dietary LA level of 1.37% led to significantly higher activities of liver lipoprotein lipase (LPL) and hepatic lipase (HL) than those of other groups (*p* < 0.05). Hepatic malate dehydrogenase (MDH) and fatty acid synthase (FAS) decreased with the increase in the dietary LA levels from 0.11% to 1.37%. The lowest MDH and FAS activities were obtained in the 1.37% LA group (*p* < 0.05). This study indicated that an appropriate amount of dietary LA was beneficial for the growth and lipid metabolism of coho salmon alevins, and the results of the quadratic regression analysis of the SGR and FCR indicated that the optimal dietary LA requirements were 1.25% and 1.23% for coho salmon alevins, respectively.

## 1. Introduction

Lipids are more digestible than carbohydrates for fish, especially for carnivorous species, such as trout and salmon [1]; they cannot synthesize the de novo n-3 or n-6 series of 18 C-polyunsaturated fatty acids (PUFAs), such as linoleic acid (18:2n-6, LA) and α-linolenic acid (18:3n-3, ALA), from oleic acid (18:1n-9) due to the lack of ∆12 and ∆15 desaturase enzymes [2,3,4,5], which need to be obtained from a dietary source [6,7,8], and that is theoretically an essential fatty acid (EFA). The two precursors (LA and ALA) were essential due to the production of longer-chained and more unsaturated counterparts (namely arachidonic acid, ARA, 20:4n-6; eicosapentaenoic acid, EPA, 20:5n-3; and docosahexaenoic acid, DHA, 22:6n-3) to meet the part of the requirement for essential highly unsaturated fatty acids (HUFAs) in fish; by the same token, a deficiency of either of them led to well-documented symptoms [2,6,9,10,11]. However, conversion efficiency hinges on the presence and expression of fatty acid (FA) desaturation and elongation genes [12], and the rate of this conversion is of paramount importance in determining the final nutritional quality of aquaculture products [13]. Previous studies have shown that lower dietary of LA than the optimal demand will lead to growth retardation on aquatic animals [14,15]. Furthermore, Chen et al.’s [16] research indicated that vegetable oil (rich in LA) that partly replaces fish oil in diets would promote the growth performance in largemouth bass. Although LA is an EFA, though one is no better than the other, excessive LA in diets possibly increases the risks of oxidative stress and weakened immune system and health [14,17,18]. In addition, although it has been confirmed that n-6 polyunsaturated fatty acids were essential fatty acids (EFAs) for normal growth and reproduction in fish [19,20,21,22,23], the exact requirements for LA and ALA widely varied between species [16]. The optimum requirement of dietary n-6 PUFAs was estimated to be about 0.5% for eel (*Anguilla japonicas*) [24], 1.0% for chum salmon (*Oncorhynchus keta*) [25], 1.0% for coho salmon (*Oncorhynchus kisutch*) [26], and 1.14% for hybrid tilapia (*Oreochromis niloticus × Oreochromis Aureus*) [27]. Compared with mammals, fish require more n-3 PUFAs than n-6 PUFAs [28,29], but studies have shown that a lack of n-6 PUFAs also significantly affects salmonids growth [25,30,31,32]. More than that, all fish species probably need a small amount of n-6 PUFAs in order to form eicosanoids [23].

The coho salmon *Oncorhynchus kisutch* (Walbaum, 1792) is one of the most important aquaculture Pacific salmon species [33], and the world produces about 120,000 tonnes of coho salmon a year from aquaculture or the wild, of which 80% of the cultured coho salmon are mainly distributed in Norway and Chile [34]. In recent years, farming of salmon has started to be promoted in China. Previous studies have shown that the EFA demand greatly varies depending on the fish species, growth stage, and environment of the fish; but most studies focus on juvenile and adult fish stages, and studies on EFAs in the early stage have not attracted enough attention [35,36,37,38]. Thus, a full understanding of the fatty acid requirements of coho salmon is particularly relevant, especially during the larval stage, since they need high-energy demands and rapid lipid metabolism, and they can be used for the commercial aquaculture of good-quality fry and fingerling [39]. Currently, EPA is characterized as EFAs [40], whereas the effects of n-6 PUFAs, especially LA, on the growth, fatty acid profile, and lipid metabolism enzyme activities of coho salmon alevins are still unknown. The present study aimed at identifying the dietary LA requirement for coho salmon alevins through evaluating the effects of graded dietary levels of LA on growth performance, fatty acid profile, and lipid metabolism enzyme activities.

## 2. Materials and Methods

### 2.1. Experimental Diets

The study was reviewed by the Experimental Animal Management Methods of Weifang University (approval number 202104132). The formulation and proximate composition and the fatty acid profile of the diets are presented in Table 1 and Table 2, respectively. Six experimental diets (containing about 47% crude protein and 15% crude lipid) were formulated to contain graded dietary LA levels (0.11%, 0.74%, 1.37%, 2.00%, 2.63%, and 3.26%) by supplementing corn oil (Shandong Conqueren Marine Technology Co., Ltd., Weifang, China) (different gradients of corn oil were filled with coconut oil to ensure the same crude lipid content in each diet group).

### 2.2. Experimental Fish and Feeding Trial

Coho salmon alevins with an initial mean body weight of 0.364 ± 0.002 g were obtained from Shandong Collaborative Innovation Center of Coho Salmon Health Culture Engineering Technology (Weifang, China). A total of 900 homogenous-sized fish were stocked in 18 white plastic tanks (0.8 × 0.6 × 0.6 m, L × W × H, water volume 240 L/tank) with 50 fish per tank in triplicate per dietary treatment. Fish were fed four times daily (7:30, 11:00, 14:30, and 18:00) to apparent satiation for 12 weeks. During the feeding period, the alevins reared in filtered underground spring water and natural light. The temperature, dissolved oxygen, and pH were in the range 15.5 ± 0.5 °C, 9.5 ± 0.8 mg L^−1^, and 6.9 ± 0.3, respectively.

### 2.3. Sampling Procedures

At the end of the feeding trial, all fish were weighed and counted after fasting for 24 h and then were anesthetized with tricaine methanesulfonate (MS-222, 20 mg L^−1^). The body total length and body, intestine somatic, and liver weights of 10 fish per replicate were recorded for the determination of the condition factor (CF), hepatosomatic index (HSI), and viscerosomatic index (VSI). Randomized sampling of 20 whole fish per replicate was performed to carry out whole-body composition. The liver and muscle samples of 15 fish in each tank were collected and stored at −80 °C until the analysis of the hepatic lipid metabolism parameters and muscular fatty acid profile.

### 2.4. Calculations and Analytical Methods

#### 2.4.1. Growth Performance2.4.2. Body Composition Analyses


$$\mathrm{Survival}\;\mathrm{rate}\;\left(\%\right)=100\times \frac{\left(\mathrm{final}\;\mathrm{amount}\;\mathrm{of}\;\mathrm{fish}\right)}{\left(\mathrm{inital}\;\mathrm{amount}\;\mathrm{of}\;\mathrm{fish}\right)}$$



$$\mathrm{Specific}\;\mathrm{growth}\;\mathrm{rate}\;\left(\mathrm{SGR},\%/\mathrm{day}\right)=100\times \frac{\left[\ln\;\left(\mathrm{final}\;\mathrm{body}\;\mathrm{weight}\right)-\ln\;\left(\mathrm{initial}\;\mathrm{body}\;\mathrm{weight}\right)\right]}{\mathrm{days}}$$



$$\mathrm{Condition}\;\mathrm{factor}\;\left(\mathrm{CF},g/{\mathrm{cm}}^{3}\right)=100\times \frac{\mathrm{body}\;\mathrm{weight}}{{\mathrm{body}\;\mathrm{length}}^{3}}$$



$$\mathrm{Hepatosomatic}\;\mathrm{index}\;\left(\mathrm{HSI},\;\%\right)=100\times \frac{\mathrm{liver}\;\mathrm{weight}}{\mathrm{body}\;\mathrm{weight}}$$



$$\mathrm{Intestine}\;\mathrm{somatic}\;\mathrm{index}\;\left(\mathrm{ISI},\%\right)=100\times \frac{\mathrm{intestine}\;\mathrm{weight}}{\mathrm{body}\;\mathrm{weight}}$$



$$\mathrm{Feed}\;\mathrm{conversion}\;\mathrm{ratio}\;\left(\mathrm{FCR}\right)=\frac{\mathrm{total}\;\mathrm{feed}\;\mathrm{intake}}{\mathrm{final}\;\mathrm{body}\;\mathrm{weight}-\mathrm{initial}\;\mathrm{body}\;\mathrm{weight}}$$


#### 2.4.2. Body Composition Analyses

The chemical compositions of diets and various tissues were analyzed following the standard methods of AOAC [41]. Moisture was determined after drying in an oven at 105 °C for 24 h. The ash content was gravimetrically determined by combustion at 550 °C for 24 h in a muffle furnace. Crude protein (N × 6.25) was determined by the Kjeldahl method. Crude lipid was determined by ether extraction using the Soxhlet method.

#### 2.4.3. Muscle Fatty Acid Analysis

FA profiles of diets and fish muscle were analyzed according to Xu et al. [42]. Briefly, the total muscle fatty acid was analyzed by a GC/MS chromatograph using Agilent Technologies 7890-5977A.

#### 2.4.4. Liver Biochemical Analysis

For the enzymatic analysis, each liver sample was homogenized in 0.1 M, pH 7.4, PBS buffer at 4 °C, and a 10% (*w/v*) homogenate was obtained. The homogenates were centrifuged for 10 min at 4 °C, and the supernatants were collected for analysis. All were estimated using commercial kits (Nanjing Jiancheng Bioengineering Institute, Nanjing, China) (including the activity of hepatic lipoprotein lipase (LPL), lipase (HL), and malate dehydrogenase (MDH) and fatty acid synthase (FAS)) by a microplate reader (Tecan Spark10M, Salzburg, Austria).

### 2.5. Statistical Analyses

Data were analyzed by one-way analysis of variance (ANOVA) in SPSS 25.0 software (Chicago, IL, USA) for Windows. Duncan’s multiple-range test was used to test the treatments’ significance. Statistical significance was set as *p* < 0.05. Regression analysis was used for analyzing the optimum LA level based on the SGR and FCR.

## 3. Results

### 3.1. Growth Performance

The survival rate ranged from 95.00% to 100.00% (Table 3), which was not significantly affected by the dietary treatments (*p* > 0.05). The final body weight and SGR were the highest in the fish fed with the diet with 1.37% of LA, which were significantly higher than those of 0.11% and 3.26% LA groups (*p* < 0.05). The FCR was the lowest in the fish fed with the diet with 1.37% of LA and significantly lower than other experimental groups (*p* < 0.05). Dietary LA levels had no significant effect on the CF, HIS, and VSI (*p* > 0.05). With the SGR as the evaluation index, the results of the quadratic regression analysis showed that the optimal dietary LA level was 1.25% (Y = −0.0882X^2^ + 0.2201X + 2.8083, R^2^ = 0.9304, *p* = 0.001) (Figure 1). With the FCR as the evaluation index, the results of the quadratic regression analysis showed that the optimal dietary LA level was 1.23% (Y = 0.1443X^2^ − 0.3551X + 1.7585, R^2^ = 0.952, *p* < 0.000) (Figure 2).

### 3.2. Whole-Body Compositions and Muscle Fatty Acid Profile

The whole-body moisture, ash, and crude protein contents were not significantly different among the dietary groups (*p* > 0.05) (Table 4). Except for the 2.00% LA group, the crude lipid content of the whole fish in the 1.37% LA group was significantly lower than that in the other groups (*p* < 0.05), and there was no significant difference among other treatment groups.

The muscle FA composition of the alevins fed with different diets is shown in Table 5. FAs 16:0 and 18:1n-9 were the most abundant FAs in all fish regardless of the different dietary LA treatments. The saturated fatty acid (SFAs) and monounsaturated fatty acid (MUFA) contents were significantly decreased (*p* < 0.05) with the increase in dietary LA levels. Both LA and ARA contents in the muscle of fish increased with an increasing dietary LA level, and the highest contents were observed in the fish fed with the diet with a 3.26% LA level. The inclusion of dietary LA significantly increased the n-6 PUFA contents, whereas no clear definite trend of the EPA and DHA contents was observed among all groups.

### 3.3. Liver Lipid Metabolic Enzymes Activities

The activities of the hepatic HL, LPL, MDH, and FAS were significantly affected by the dietary LA levels. The HL and LPL activities were found increasing until the 1.37% group, after which a reduction was noticed from the 2.00% LA group, and the rest of the HL and LPL activities were showing a fluctuating trend (*p* < 0.05) (Figure 3A,B). On the contrary, the MDH activity was found significantly decreasing with the dietary LA level from 0.11% to 1.37% (*p* < 0.05) and then increased with the dietary LA level increasing from 2.00% to 3.26% (Figure 3C). Similarly, the 1.37% LA group also had the lowest FAS activity among all groups, except for the 0.11% and 3.26% LA groups, which had no significant differences from the other groups (Figure 3D).

## 4. Discussion

Long-term absence of EFAs (including LA, ALA, EPA, and DHA) from the diet leads to deficiency signs that most often include reduced growth and increased mortality [43]. In the present study, there were no significant differences in the survival rates among all groups; however, the relatively low survival rates of 0.11% and 0.74% LA groups (95.00% and 97.50%, respectively) seemed to reflect that dietary LA deficiency affected the survival rate of coho salmon alevins. This phenomenon may confirm that LA is an essential fatty acid for coho salmon alevins, and an appropriate supplemental level is beneficial to the survival rate of fish. According to the regression relationship between dietary LA contents and SGR and FCR, the optimum dietary LA contents of coho salmon alevins were 1.23% and 1.25%, respectively. The results showed that the growth of alevins was greatly affected by the dietary LA levels and could be remarkably enhanced by the diets supplemented with appropriate LA, which was in agreement with previous studies in chum salmon [25], rainbow trout [30], Russian sturgeon (*Acipenser gueldenstaedti*) [19], and tilapia [27]. Stickney et al. [44] found that the growth of *T. aurea* was improved by diets containing high soybean oil levels (which are fats with linoleic acid). Kanazawa et al. [45] found that 1.0% LA in the feed was the most suitable for tilapia growth. However, Takeuchi et al. [46] found that dietary 0.5% LA was the optimal requirement for Nile tilapia. However, when the diet was supplemented with excessive LA (LA > 1.37%), the growth performance and conversion efficiency of coho salmon showed a downward trend, which may be due to the deficiency of n-3 fatty acids in diets; thus, it might affect fish growth and feed utilization [47,48]. These results indicate that it is essential to add appropriate LA to the diet; however, there are slight differences between fish species. Hence, only the appropriate content of essential fatty acids can ensure the better growth performance and feed utilization efficiency of fish.

The present study showed that the CF, HSI, and VSI are not affected by the dietary LA contents. Notably, the HSI was introduced as an indicator to estimate lipid reserves in the aquatic [49,50], which is normally used to indicate the effect of the feeding regime on liver functionality [51]. Similar results have been reported for other fish [13,52,53,54]. These results might indicate that the dietary LA level itself does not directly cause the lipid deposited in the liver. Rather, it has been reported that high lipid or highly unsaturated oil inclusion in the diet may be responsible for the increase in liver weight [55,56,57].

In the present study, the fatty acid composition of coho salmon alevins generally reflected their dietary composition, as reported in many other studies [6,52,58,59,60,61,62]. Similarly, the content of unsaturated FA in the fish body would be affected by the unsaturated oils in the diet. In the study of Russian sturgeon (*Acipenser gueldenstaedti*), using a diet with coconut oil, and with the coconut oil in the basal diet being replaced by sunflower oil, the total n-6 fatty acid content in the whole body showed the same trend as the dietary sunflower oil content [19]. It was confirmed in this study that the group with the highest dietary LA content had the highest C18:2n-6 and total n-6 fatty acid contents in the muscle. Similar results were also found in Nile tilapia (*Oreochromis niloticus*) fry and yellow catfish (*Pelteobagrus fulvidraco*) [63,64]. Koven et al. [65,66] found that diets enriched with C20:4n-6 (ARA) converting from LA were beneficial to the growth and overall survival rate of larval gilthead sea bream. Significantly, the content of ARA in the muscle of fish also increased with the dietary LA level increasing, suggesting that the LA intake of coho salmon alevins could synthesize ARA through the desaturation of carbon chain extension. However, the muscle ALA level was not affected by dietary LA, which may be due to the fact that the dietary ALA level in each group was added in a consistent amount. On the other hand, since LA and ALA serve as the substrates for desaturase and elongase enzymes in the biosynthesis pathways (the bioconversion of LA to ARA and ALA to EPA/DHA) [23,37], the muscle ARA, EPA, and DHA (which were converted from LA and ALA) are thus substantially affected by the dietary LA level. However, the results of this experiment showed that muscle ARA, EPA, and DHA were not significantly affected by the dietary LA level. This is possibly caused by the addition of fish oil to the diet, which provided sufficient HUFA for alevins.

Lipoprotein lipase (LPL) and hepatic lipase (HL) are key enzymes in liver lipid catabolism, and both catalyze the hydrolysis of triglyceride (TG) in chylomicron (CM) and very low-density lipoprotein (VLDL), releasing free fatty acids (FFAs) for storage in lipid storage organs or for oxidation in other organs. The HL can also be used as a ligand to promote the uptake of cholesterol (CHO) or VLDL residues in high-density lipoprotein (HDL) by hepatocytes [67,68,69,70]. MDH produces NADPH and NADH that are essential for lipid synthesis [19]. There is mounting evidence that the dietary lipid levels and fatty acids type have an impact on lipid metabolic enzymes in fish; in addition, the imbalanced n-3 and n-6 fatty acids in the dietary can generally inhibit the activity of metabolism-related enzymes [19,71,72]. Higher dietary lipid content resulted in elevated activities of LPL and HL during the early ontogeny of *Pelteobagrus vachelli* larvae [73]. A similar result was also reported by Li et al. [19] who found that the activities of LPL and lipase (LPS) in the liver were markedly increased by the increase in dietary n-6 fatty acids (1.0% LA group) for Russian sturgeon. In this study, fish fed with the diet with a 1.37% LA level had the lowest deposition of lipid in the whole body. Compared with the 1.37% LA diet group, feeding the alevins with higher or lower levels of LA led to the increase in whole-body lipid deposition, which might be attributed to the lack of LA or the suboptimal dietary EFA content. In addition, the diet with LA of 1.37% level led to the lowest activity of MDH in the liver, confirming that the appropriate dietary LA supplementation ratio could reduce the lipid synthesis in vivo, thereby improving the lipid decomposition and transport and inhibiting the lipid deposition in coho salmon alevins. Menoyo et al. [74] found that dietary PUFAs (n-3/n-6 ratios at 4.7 and 5.4) were negatively correlated with MDH in the liver of Atlantic salmon. Li et al. [19] also found that compared with the 2.0% LA group, the 1.0% LA group had higher activity of LPL in the liver of Russian sturgeon, whereas the MDH activity was significantly lower. The present study indicated that the dietary FA composition affected the growth and utilization of lipid, and the suitable supplementation of n-6 fatty acids (especially LA) would be conducive to the activity of LPL and HL, reduce lipid synthesis, and increase lipid decomposition and transport, but negatively affected the activity of MDH to prevent excessive lipid deposition.

As a key enzyme of lipogenesis, FAS plays a crucial role in the weight variability of the abdominal adipose tissue [75] and is mainly expressed in the hepatic cells of fish [76]. The activity and mRNA expression of FAS are regulated by dietary fatty acids [77,78,79] and hormones [80]. FAS activity in fish decreases with the increase in the dietary lipid content, thus inhibiting the production of lipid [81]. Dai et al. [14] found that the dietary LA level had no significant effect on the FAS expression of Chinese mitten crabs (*Eriocheir sinensis*); however, the FAS expression level was the lowest in the recommended addition range of dietary LA (19.77 g to 28 g/kg). A different result was reported in turbot (*Scophthalmus maximus* L.) and silver pomfret (*Pampus argenteus*) [82,83] in which hepatic FAS expression significantly upregulated with 100% soybean oil (rich in n-6 PUFA) replacement of fish oil in diets, indicating that FAS was upregulated by high dietary n-6 PUFA levels. This may be attributed to the addition of fish oil (containing EPA and DHA) in this experiment, in which two fatty acids inhibited the transcription and translation of FAS [84]. More than that, previous research found that high dietary EPA content significantly reduced the hepatic FAS activity of alevins [40].

## 5. Conclusions

In summary, the different dietary levels of fatty acids significantly affected the growth, SGR, FCR, and activities of hepatic lipid metabolic enzymes (HL, LPL, MDH, and FAS) in coho salmon *Oncorhynchus kisutch* alevins. LA is essential for growth and lipid utilization of coho salmon alevins. Based on the SGR and FCR, the dietary LA requirements were estimated to be 1.25% and 1.23% for coho salmon alevins, respectively. Here, we recommend that the dietary LA in coho salmon alevins should fluctuate from 0.74% to 1.37%. This result helps us to provide a theoretical basis for the use of LA in the coho salmon diet and then provides guidance for practical production. Further studies are required to determine the optimum proportion of different fatty acids (LA and ALA), and the effects of dietary fatty acid (particularly LA or other n-6 HUFAs) levels on the molecular mechanism of coho salmon lipid metabolic need to be further explored.

## Figures and Tables

**Figure 1 animals-12-02631-f001:**
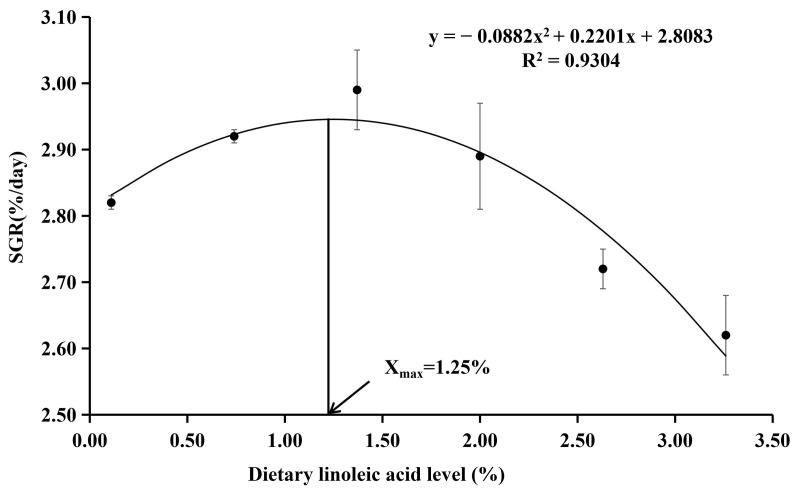
Quadratic regression analysis of SGR with dietary linoleic acid (LA) levels in coho salmon *Oncorhynchus kisutch* alevins. Estimated dietary LA requirement of SGR is 1.25%. Each point represents the mean of three replicates.

**Figure 2 animals-12-02631-f002:**
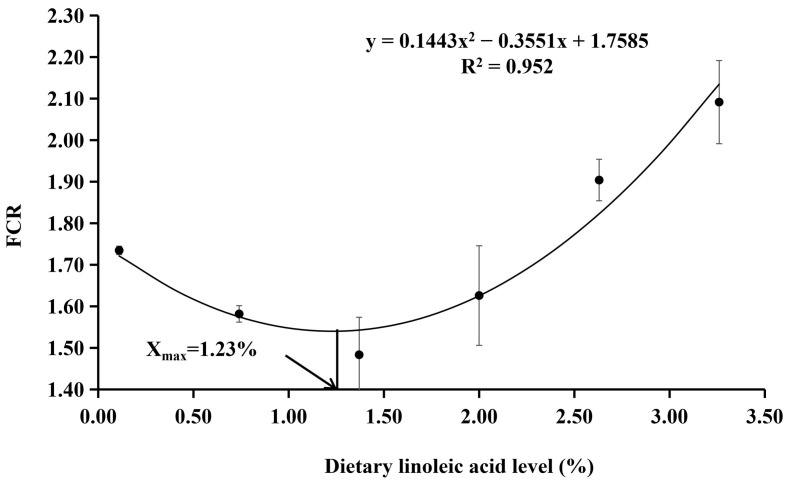
Quadratic regression analysis of FCR with dietary LA levels in coho salmon *Oncorhynchus kisutch* alevins. Estimated dietary LA requirement of SGR is 1.23%. Each point represents the mean of three replicates.

**Figure 3 animals-12-02631-f003:**
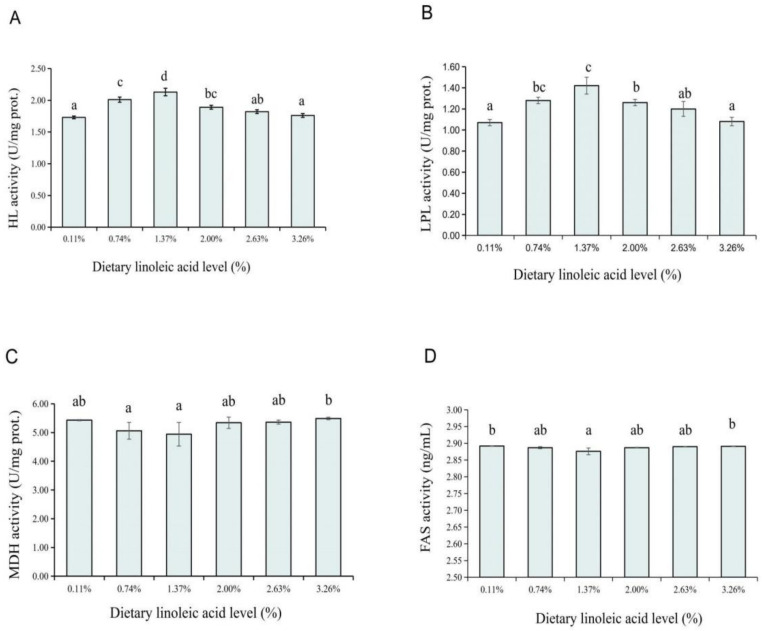
Effects of dietary LA levels on HL (hepatic lipase, (**A**)), LPL (lipoprotein lipase, (**B**)), MDH (malate dehydrogenase, (**C**)), and FAS (fatty acid synthase, (**D**)) activities in liver of coho salmon *Oncorhynchus kisutch* alevins. ^a–d^: Means that do not share similar letter in group are significantly different, *p* < 0.05.

**Table 1 animals-12-02631-t001:** Formulation and proximate composition of experimental diets for coho salmon *Oncorhynchus kisutch* alevins (% in dry matter).

Ingredients (%)	Dietary Linoleic Acid Level (%)					
	0.11	0.74	1.37	2.00	2.63	3.26
Degreasing fish meal ^1^	40.00	40.00	40.00	40.00	40.00	40.00
Soybean protein concentrate ^1^	5.00	5.00	5.00	5.00	5.00	5.00
Soybean meal ^1^	20.00	20.00	20.00	20.00	20.00	20.00
Peanut meal ^1^	9.80	9.80	9.80	9.80	9.80	9.80
α-Starch ^1^	12.00	12.00	12.00	12.00	12.00	12.00
Sodium alginate ^1^	2.00	2.00	2.00	2.00	2.00	2.00
Soybean lecithin ^1^	1.80	1.80	1.80	1.80	1.80	1.80
Fish oil ^1^	2.00	2.00	2.00	2.00	2.00	2.00
Coconut oil ^1^	6.00	4.80	3.60	2.40	1.20	0.00
Corn oil	0.00	1.20	2.40	3.60	4.80	6.00
Mineral premix ^2^	0.50	0.50	0.50	0.50	0.50	0.50
Vitamin premix ^3^	0.50	0.50	0.50	0.50	0.50	0.50
Vitamin C phosphate	0.05	0.05	0.05	0.05	0.05	0.05
Choline chloride	0.30	0.30	0.30	0.30	0.30	0.30
Ethoxyquin	0.05	0.05	0.05	0.05	0.05	0.05
Proximate composition						
Moisture (%)	7.33	7.29	7.60	7.20	7.32	7.38
Crude protein (%)	46.65	46.90	46.98	46.73	47.00	47.01
Crude lipid (%)	13.67	14.16	13.79	13.94	14.01	13.84
Ash (%)	9.11	9.29	9.38	9.39	9.27	9.47

^1^ Provided by Shandong Conqueren Marine Technology Co., Ltd., Weifang, China. ^2^ Composition (mg kg^−1^ mineral premix): AlK (SO_4_)_2_·12H_2_O, 123.7; CaCl_2,_ 17,879.8; CuSO_4_·5H_2_O, 31.7; CoCl_2_·6H_2_O, 48.9; FeSO_4_·7H_2_O, 707.4; MgSO_4_·7H_2_O, 4316.8; MnSO_4_·4H_2_O, 31.1; ZnSO4·7H2O, 176.7, KCl, 1191.9; KI, 5.3; NaCl, 4934.5; Na_2_SeO_3_·H_2_O, 3.4; Ca (H_2_PO_4_)_2_·H_2_O, 12,457.0; KH_2_PO_4,_ 9930.2. ^3^ Composition (IU or g kg^−1^ vitamin premix): retinal palmitate, 10,000 IU; cholecalciferol, 4000 IU; *α*-tocopherol, 75.0 IU; menadione, 22.0 g; thiamine-HCl, 40.0 g; riboflavin, 30.0 g; D-calcium pantothenate, 150.0 g; pyridoxine-HCl, 20.0 g; meso-inositol, 500.0 g; D-biotin, 1.0 g; folic acid, 15.0 g; ascorbic acid, 200.0 g; niacin, 300.0 g; cyanocobalamin, 0.3 g.

**Table 2 animals-12-02631-t002:** Fatty acid composition dry matter of lipid ingredient (% total FA) and experimental diets (%).

Fatty Acids	Coconut Oil	Corn Oil	Dietary LA Level (%)					
			0.11	0.74	1.37	2.00	2.63	3.26
C12:0	28.34	0.02	0.783	0.627	0.471	0.315	0.159	0.002
C14:0	15.22	0.01	0.317	0.254	0.191	0.127	0.064	0.001
C16:0	13.28	10.90	0.252	0.242	0.232	0.221	0.211	0.201
C16:1n-7	0.02	0.11	0.004	0.004	0.004	0.004	0.004	0.004
C18:0	3.46	1.81	0.026	0.022	0.019	0.016	0.012	0.009
C18:1n-9	2.13	25.2	0.023	0.164	0.304	0.445	0.586	0.727
C18:3n-3(ALA)	0.83	1.35	0.007	0.007	0.008	0.009	0.009	0.010
C20:5n-3(EPA)	-	-	0.002	0.002	0.002	0.002	0.002	0.002
C22:6n-3(DHA)	-	-	0.002	0.002	0.002	0.002	0.002	0.002
C20:0	1.23	0.20	0.014	0.012	0.010	0.008	0.006	0.004
C20:1	1.03	0.21	0.013	0.011	0.010	0.008	0.006	0.004
C18:2n-6(LA)	1.55	54.4	0.110	0.740	1.370	2.000	2.630	3.260
C20:4n-6(ARA)	0.05	-	0.001	0.001	0.001	0.001	0.001	0.001
∑n-6PUFA	12.36	55.2	0.214	0.835	1.420	2.006	2.962	3.631

-: Means that not detected.

**Table 3 animals-12-02631-t003:** Survival, growth performance, and feed utilization of *Oncorhynchus kisutch* alevins fed with experimental diets with different dietary linoleic acid levels for 12 weeks (means ± SE, *n* = 3).

Dietary LA Level (%)	0.11	0.74	1.37	2.00	2.63	3.26	*p* Value
Survival rate (%)	95.00 ± 2.89	97.50 ± 2.50	100.00 ± 0.00	100.00 ± 0.00	100.00 ± 0.00	100.00 ± 0.00	0.146
Initial body weight (g)	0.365 ± 0.001	0.362 ± 0.002	0.363 ± 0.001	0.364 ± 0.001	0.364 ± 0.002	0.366 ± 0.002	0.609
Final body weight (g)	3.86 ± 0.02 ^b^	4.24 ± 0.04 ^c^	4.46 ± 0.05 ^d^	4.07 ± 0.07 ^c^	3.54 ± 0.10 ^a,b^	3.27 ± 0.07 ^a^	0.011
SGR (% day^−1^)	2.82 ± 0.01 ^b^	2.92 ± 0.01 ^c^	2.99 ± 0.06 ^d^	2.89 ± 0.08 ^c^	2.72 ± 0.03 ^a,b^	2.62 ± 0.06 ^a^	0.010
FCR	1.73 ± 0.01 ^c^	1.58 ± 0.02 ^b^	1.48± 0.01 ^a^	1.63 ± 0.02 ^b^	1.90 ± 0.03 ^c,d^	2.09 ± 0.04 ^d^	0.010
CF	1.27 ± 0.11	1.13 ± 0.17	1.02 ± 0.03	1.17 ± 0.10	1.04 ± 0.09	1.11 ± 0.02	0.582
HSI	1.27 ± 0.03	1.27 ± 0.01	1.22 ± 0.02	1.27 ± 0.05	1.26 ± 0.07	1.36 ± 0.14	0.790
VSI	1.73 ± 0.07	1.72 ± 0.05	1.51 ± 0.20	1.43 ± 0.07	1.50 ± 0.02	1.46 ± 0.04	0.194

Means in the same raw with different superscript letters are significantly different (*p* < 0.05).

**Table 4 animals-12-02631-t004:** Whole-body proximate composition of coho salmon *Oncorhynchus kisutch* alevins fed with experimental diets with different dietary LA levels for 12 weeks (means ± SE, *n* = 3).

Dietary Linoleic Acid Level (%)	Moisture (%)	Crude Protein (%)	Crude Lipid (%)	Ash (%)
0.11	77.89 ± 0.13	12.63 ± 0.17	4.40 ± 0.01 ^b^	3.64 ± 0.03
0.74	77.46 ± 0.01	12.99 ± 0.20	4.36 ± 0.61 ^b^	3.55 ± 0.01
1.37	77.91 ± 0.24	12.71 ± 0.33	4.12 ± 0.59 ^a^	3.45 ± 0.03
2.00	77.98 ± 0.06	12.23 ± 0.10	4.24 ± 0.30 ^ab^	3.45 ± 0.06
2.63	77.44 ± 0.27	13.11 ± 0.30	4.55 ± 0.33 ^b^	3.60 ± 0.02
3.26	77.62 ± 0.21	12.78 ± 0.37	4.53 ± 0.31 ^b^	3.65 ± 0.03
One-way ANOVA				
*p* value	0.293	0.843	0.049	0.206

Means in the same column with different superscript letters are significantly different (*p* < 0.05).

**Table 5 animals-12-02631-t005:** Muscle fatty acid composition (% Total FA) of *Oncorhynchus kisutch* alevins fed with experimental diets with different dietary linoleic acid levels for 12 weeks (on dry matter basis) (means ± SE, *n* = 3).

Fatty Acids	Dietary LA Level (%)					
	0.11	0.74	1.37	2.00	2.63	3.26
C14:0	2.24 ± 0.03 ^d^	2.23 ± 0.01 ^d^	2.22 ± 0.02 ^d^	1.88 ± 0.02 ^c^	1.77 ± 0.03 ^b^	1.56 ± 0.02 ^a^
C16:0	22.02 ± 0.24 ^f^	20.6 ± 0.17 ^e^	18.44 ± 0.06 ^d^	16.06 ± 0.17 ^c^	14.52 ± 0.08 ^b^	12.51 ± 0.03 ^a^
C18:0	6.09 ± 0.04 ^b^	5.68 ± 0.16 ^a,b^	5.56 ± 0.06 ^a^	5.37 ± 0.25 ^a^	5.28 ± 0.16 ^a^	5.25 ± 0.06 ^a^
∑SFA	30.34 ± 0.27 ^f^	28.53 ± 0.31 ^e^	26.22 ± 0.02 ^d^	23.3 ± 0.44 ^c^	21.58 ± 0.16 ^b^	19.32 ± 0.10 ^a^
C16:1n-7	7.27 ± 0.10 ^e^	7.14 ± 0.06 ^e^	6.67 ± 0.13 ^d^	5.47 ± 0.82 ^c^	4.23 ± 0.04 ^b^	3.33 ± 0.04 ^a^
C18:1n-9	33.58 ± 0.11	32.54 ± 0.62	31.53 ± 0.27	32.6 ± 0.20	33.54 ± 0.03	31.47 ± 0.22
C20:1	1.17 ± 0.03 ^c^	1.14 ± 0.06 ^c^	0.86 ± 0.01 ^b^	0.82 ± 0.01 ^a,b^	0.76 ± 0.02 ^a^	0.75 ± 0.02 ^a^
∑MUFA	42.02 ± 0.20 ^c^	40.82 ± 0.74 ^b,c^	39.06 ± 0.39 ^b^	38.89 ± 0.28 ^b^	38.52 ± 0.08 ^b^	35.55 ± 0.27 ^a^
C18:3n-3(ALA)	0.43 ± 0.012	0.42 ± 0.005	0.46 ± 0.005	0.41 ± 0.006	0.47 ± 0.005	0.43 ± 0.006
C20:5n-3(EPA)	0.80 ± 0.005	0.75 ± 0.005	0.76 ± 0.006	0.82 ± 0.005	0.81 ± 0.006	0.80 ± 0.008
C22:6n-3(DHA)	2.23 ± 0.01	2.22 ± 0.01	2.18 ± 0.03	2.20 ± 0.01	2.22 ± 0.01	2.22 ± 0.03
∑n-3 PUFA	3.26 ± 0.02	3.19 ± 0.01	3.25 ± 0.03	3.39 ± 0.02	3.46 ± 0.01	3.51 ± 0.02
C18:2n-6 (LA)	4.92 ± 0.02 ^a^	6.08 ± 0.03 ^b^	10.38 ± 0.04 ^c^	15.6 ± 0.02 ^d^	19.64 ± 0.01 ^e^	23.46 ± 0.20 ^f^
C18:3n-6	0.49 ±0.003 ^a,b^	0.51 ± 0.03 ^c,d^	0.51 ± 0.003 ^d^	0.49 ± 0.008 ^b,c^	0.49 ± 0.008 ^b,c^	0.47 ± 0.003 ^a^
C20:4n-6(ARA)	2.05 ± 0.015 ^a^	2.08 ± 0.006 ^b^	2.11 ± 0.003 ^c^	2.12 ± 0.008 ^c^	2.14 ± 0.006 ^c^	2.17 ± 0.006 ^d^
∑n-6 PUFA	5.54 ± 0.05 ^a^	8.19 ± 0.01 ^b^	12.61 ± 0.01 ^c^	18.08 ± 0.02 ^d^	23.21 ± 0.01 ^e^	26.28 ± 0.02 ^f^

Means in the same raw with different superscript letters are significantly different (*p* < 0.05).

## Data Availability

All the data in the article are available from the corresponding author upon reasonable request.

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
