# Peer review of "Effect of Dietary Linoleic Acid (18:2n-6) Supplementation on the Growth Performance, Fatty Acid Profile, and Lipid Metabolism Enzyme Activities of Coho Salmon (Oncorhynchus kisutch) Alevins"

_animals, 2022, doi:10.3390/ani12192631_

Round 1
Reviewer 1 Report
Overall, I feel that a major revision is required before the work meets the standard for publication in the journal. The manuscript should also be corrected for English as some sentences/words have not been appropriately presented.
I have the following high-level comments:
Line 22: “linoleic acid was used as the only variableis”, this is not the case in the present study as other fatty acids varied across diets
Line 51-83: I strongly recommend the authors provide more data on Coho salmon production and distribution, among others, to further convince the readers about the importance of the present study. It is also recommended that the need for investigating proper dietary LA for this species should be highlighted
Line 84: The authors are recommended to include an ethical statement in this section
Line 92: The authors should specify the unit and form for “Formulation” in % as it or % dry matter
Line 102: The fatty acid profile of corn oil and coconut oil should be presented
Line 102: “Ingredients” should be removed from the first line of the Table
Line 102: Since the authors mentioned the dietary level of LA in %, the Table 2 should be presented in % as it or % dry matter for all fatty acids
Line 104: Please check the decimal place of the mean initial weight, which should be aligned with that of the standard deviation
Line 112-120: Were feed deprived for fish before the biometrics and sampling?
Line 118: Fish muscle sampling was not mentioned here. Why?
Line 131: For how long?
Line 132: For how long?
Line 137: total lipid can not be analyzed by such a device? The authors meant fatty acids?.
Line 160: Once the authors mention the regression model, it is suggested to indicate the P-value of the test along with R-square, similar comment for Line 162
Line 171-172: The lipid did not differ in these groups compared to 0.11% and 3.26%? Please revise the statement
Line 184: Please specify the muscle in the title of table 2
Line 184-185: Ingredients should be removed from the headline of the table
Line 191: Table 3 was not related to these results
Please check the rest of the manuscript for such errors
Minor comments:
Line 17, 18: Please check the proper words “particularlyrelevant”, “rapidlipid”
Line 19: “a” not “an”
Line 37-38: Please check the grammar for this sentence
Line 86-87: presented NOT present
Line 107: Space is needed between “)” and “with”
Reviewer 2 Report
The manuscript in question deals with identifying the dietary LA requirement for coho salmon alevins through evaluating the effects of graded dietary levels of LA on the growth performance, fatty acid profile and lipid metabolism enzyme activities. The area of ​​fish nutrition in aquaculture is essential for the production of better quality and greater profitability for breeders. This research used an important marine fish species that represents recent activity in the region. As it is a recent production, studies must be carried out to verify the best feed for these fish species in their new environment. The article is well written, needs a slight linguistic revision and there are some points that should be revised.
- consider the most recent way of writing the units, such as mg/L now which is mg L-1 and others used in the text, please correct them.
- in Material and Methods it was not clear the replacement of corn oil by coconut oil. In Table 1 there is still corn oil. In fact, the feed contains both sources of lipids: corn and coconut, right? Please clarify.
- CF, HSI and VSI were calculated, with results presented but not discussed. While there were no significant differences between the diets for these indices, they do mean something and should be discussed. Enzymes also need to be further discussed.
- Figures 2 and 3 are out of context and Figure 3 has no caption.
- on lines 262 3 263 "thereby" is repeated
- some scientific names, in the references, need to be written in italics.
- Conclusions could be more direct and elucidative, specifying which concentrations are best for the general performance of the fish
Reviewer 3 Report
The manuscript has some merits and covers an issue related the alevins fatty acids nutrition that was previously poorly explored. A number of corrections and improvements in the text are needed. Line 18: ...rapid lipid...; line 19: ...a vital...;line 23: .....enzymes; line 28: A 12-weeks...; lines 58-62 not clear: do actually the author mean that only LA and ALA are essential in the diet and that EPA as well as DHA are sufficiently synthesized by the fish? The titles of the figures should be reported below. Table 2: EPA and DHA are missed; line 115: standard or total length? line 202: Discussion should be moved after the title of figure 3, that should be placed below the figure. Line 213: not true over 1.37%. Line 328:....consistent amount. On the other hand,...Lines 239-243: not clear, a better explication needed.
Coherence: in line 55 and elsewhere the C18:2n-6 is indicated as LA, while on table 2 is reported as LNA: authors should correct in LA
Moreover: The reduced growth and conversion efficiency for LA > 1.37% should be better interpreted. Was it due to a negative effect of LA or to a reduction of other FAs operated to maintain almost equals the crude lipids in tall the diets?
MDH and FAS dynamics for LA < and> of 1.37% should be better explained
Therefore the manuscript needs some more than minor revisions by the authors.
Round 2
Reviewer 1 Report
In general, the authors have clarified most of my concerns/comments. However, some critical notes are held as follows:
- The whole manuscript should be undergone English correction
- I am still not satisfied with the coherence made in the introduction part. I strongly recommend the authors to work on this part to provide deeply to provide robustness and attraction to the readers
- Table 2:
+ Fatty acid profile should be presented as "% as fed" (as it) NOT "% total FA"
+ Since the only variation across the experimental diets was the substitution of coconut oil for corn oil, why was there a significant difference in C16:0 across diets, even though both oil sources provided equivalent amounts of this fatty acid?
- Line 179 and 181-182: Is it weird that the authors provided two values for R-square for one regression model, and those values were substantially different (might it be adjusted R-square?). Please check the regression/correlation analysis outputs and pick the correct values.
- Please revise all syntax in the manuscript. For instance, Line 164: there should be a space before..."by a..), and so on
